# Estimation of the Mutagenic Potential of 8-Oxog in Nuclear Extracts of Mouse Cells Using the “Framed Mirror” Method

**DOI:** 10.3390/mps3010003

**Published:** 2020-01-03

**Authors:** Leonid V. Gening, Alexandr A. Volodin, Konstantin Y. Kazachenko, Irina V. Makarova, Vyacheslav Z. Tarantul

**Affiliations:** Institute of Molecular Genetics, Kurchatov Sq. 2, Moscow 123182, Russiakonstantinkazach@yandex.ru (K.Y.K.); ivmakarova@img.ras.ru (I.V.M.); tarantul@img.ras.ru (V.Z.T.)

**Keywords:** DNA lesions, 7,8-dihydro-8-oxoguanine (8-oxoG), mutagenic activity, method of detection, cell nuclear extracts, mice organs and embryos

## Abstract

We propose an improved earlier described “mirror” method for detecting in cell nuclear extracts mutations that arise in DNA during its replication due to the misincorporation of deoxyadenosine-5′-monophosphate (dAMP) opposite 7,8-dihydro-8-oxoguanine (8-oxoG). This method is based on the synthesis of a complementary chain (“mirror”) by nuclear extracts of different mice organs on a template containing 8-oxoG and dideoxycytidine residue (ddC) at the 3′‑end. The “mirror” was amplified by PCR using primers part of which was non-complementary to the template. It allowed obtaining the “framed mirror” products. The misincorporation of dAMP in “framed mirror” products forms an *EcoRI* restriction site. The restriction analysis of double-stranded “framed mirror” products allows a quantification of the mutation frequency in nuclear extracts. The data obtained show that the mutagenic potential of 8-oxoG markedly varied in different organs of adult mice and embryos.

## 1. Introduction

One of the main causes of GC-AT transversion somatic mutations is a DNA lesion, 8-oxoguanine (8-oxoG) arising in the genome under the action of reactive oxygen species (ROS) generated in cells in the processes of oxidative phosphorylation [1,2]. The intensity of the conversion of this DNA lesion into a mutation is a key moment of aging and oncological, neurological, and some other diseases [3,4]. The presence of 8-oxoG is dangerous for the stability of the genome because all known DNA polymerases insert, with different frequencies, a correct dC or an incorrect dA opposite this lesion. For replicative DNA polymerases, the insertion of dA is close to 100% [5]. These data show that mutations owing to 8-oxoG arise in the process of DNA replication, mostly due to the appearance of mutant dA residues in the nascent DNA, instead of correct dC [6].

In mammalian cells, there is a base excision repair (BER) pathways that reduce the mutational burden of ROS, ensuring the correct and efficient repair of A: 8-oxo G mispairs and the removal of 8-oxoG lesions from the genome [3]. In 8-oxoG:dA mispairs, dA is removed by MutYH glycosylase and DNA polymerase lambda for post-replicative repair [7,8].

The mutagenic potential of 8-oxoG was studied over decades by various authors using different experimental systems. As early as 1991, Shibutani et al. [5] showed that the ratio of incorporated dA to dC opposite 8-oxoG varied for different DNA polymerases. These authors conducted the reaction of primer extension by purified DNA polymerases on an artificial template containing 8-oxoG. The copies obtained were separated by electrophoresis in denaturing polyacrylamide gel. It allowed to separate DNA molecules with a definite mutation from those lacking this mutation, whereas DNA with other mutations that may appear in the experiment was absent. Although the advantages of this method were obvious, it was not used or improved over the last 28 years. The reason for this was probably that the separation of equal-sized DNA molecules that differed only in one nucleotide is technically complicated.

In later experiments, a technically less complicated approach was used, whereby a primer extension assay with four parallel DNA polymerase reactions was carried out simultaneously (each of them with only one of four dNTPs) was usually performed to ascertain which nucleotide is incorporated opposite a certain lesion [9,10,11]. However, in all these experiments, the primer extension reaction products were not separated by electrophoresis, and therefore no conclusion could be drawn concerning the question if these lesions cause the misincorporation of only one type of nucleotides and don’t generate other mutations as well.

The most advanced is the method where, at first, a specific DNA polymerase is used to obtain a copy of the template containing a DNA lesion. The obtained copy is then sequenced and studied for mutations [12]. This method allows for identifying nucleotides incorporated in the copy opposite this or that DNA lesion, and to determine if there are closely located other mutations.

Recently we proposed a “mirror” method to study the mutagenic potential of 8-oxoG using purified DNA polymerases [1]. At the first stage of this method, similar to the method used by Targgart et al. [12], a specific DNA polymerase is used to obtain a copy of the template that we called “mirror”. The “mirror” method differs in that, instead of sequencing “mirror”, the nucleotide opposite the lesion is identified by simple restriction analysis, whereas other mutations can be identified by the Single strand conformation polymorphism (SSCP) method. The “mirror” method allows for the analyses of much more material in a short time without additional sequencing.

However, studies with purified DNA polymerases give only a partial idea of the real situation in vivo. A more detailed information about the mutagenic potential of 8-oxoG in normal cells and in various pathologies can be obtained from using cell extracts containing the whole set of DNA polymerases and regulatory proteins. Such an approach facilitated determining the role in the correction of DNA synthesis of not only DNA polymerases, but also other proteins involved in DNA replication. For example, experiments using model DNA template, purified DNA polymerases and extracts of knocked down cells showed that accessory replication protein A and PCNA function as molecular switches that activated the efficient incorporation of correct dC opposite 8-oxoG of the template by DNA polymerase lambda, but blocked the incorporation of incorrect (mutant) dA [13].

Thus, on the one hand, certain DNA polymerases can, with a high probability, incorporate mutant dA opposite 8-oxoG in the template, and on the other hand, certain regulatory proteins inhibit this and support the trend to incorporate correct dC. However, in contrast to purified DNA polymerases, nuclear and cell extracts of mammalian cells contain not only the full set of enzymes involved in DNA synthesis and repair, but also many nucleases. These nucleases partially destroy DNA and thereby form its shorter fragments. As a result, the synthesized DNA fragments have different lengths, and only part of them can correspond to the full-size copy of the template (“mirror”). All of this complicates the use of the “mirror” method with cell extracts.

In the present work, we modified the “mirror” method and called it the “framed mirror” method. This new method allows to overcome the existing experimental difficulties and to determine the mutagenic potential of 8-oxoG in nuclear extracts of different mouse cell organs. The results obtained show that the bypass of DNA lesions caused by 8-oxoG is different in different organs of adult animals and in embryos.

## 2. Materials and Methods

### 2.1. Mice and Embryos

The animals were treated in accordance with the European Society Council 86/609/EEC Requirement concerning the use of animals for experimental studies. The extracts of mouse organs were obtained from C57B1 strain mice (about 3 months of age). Mouse embryos were isolated on day 12–16 of the fetal development.

### 2.2. Preparation of Nuclear Extracts

Cell nuclear extracts were prepared according to a slightly modified method described by Schreiber et al. [14]. All procedures were conducted at 0 °C. Typically, 300 mg of a mouse organ was transferred into an Eppendorf tube filled with an equal volume of 20 mM Tris-HCl buffer, pH 7.5, with 1 mM EDTA, 1 mM phenylmethylsulfonylfluoride (PMSF), and 1 mM DTT. The content of the tube was then homogenized with a Teflon pestle for 5 min. The homogenate was then centrifuged for 5 min at 14,000 rpm in an Eppendorf centrifuge. The pellet was resuspended in an equal volume of 20 mM Tris-HCl pH 7.5, 60 mM KCl, 1 mM EDTA, 1 mM PMSF and 1 mM DTT. Then it was homogenized for 5 min with a Teflon pestle and centrifuged for 5 min at 14,000 rpm. The pellet was resuspended in an equal volume of 20 mM Tris-HCl pH 7.5, with 450 mM NaCl, 1.5 mM MgCl_2_, 0.2 mM EDTA, 1 mM PMSF, and 25% glycerol, and homogenized one more time. The homogenate obtained was incubated for 15 min at 0 °C with stirring, and then centrifuged as described above. The last stage was repeated twice. Two fractions obtained by extraction in buffer with high salt concentration (450 mM NaCl) were poured together and dialyzed for 5 h against 300 mL Tris‑HCl buffer pH 7.5, 100 mM KCl, 0.2 mM EDTA, 0.5 mM DTT and 0.1 mM PMSF. Aliquots of the dialysate were stored at −70 °C. Protein concentration in the fractions (10–15 mg/mL) was measured by the Bradford method [15].

### 2.3. Templates and Primers for “Framed Mirror” Product Synthesis

For the work, we used oligonucleotides synthesized by the firm “DNA synthesis” and “Eurogene”, Moscow.

Templates:

AXT/ddC (with 8-oxoG (X) and dideoxycytidine (ddC)):

5′-GGGATCCTGCTGCCATAGGAA**X**TCTTGATTGGAAAGTCGACCTGddC-3′.

45(T) (with dT in the underlined *EcoRI* restriction site):

5′-GGGATCCTGCTGCCATAGGAA**T**TCTTGATTGGAAAGTCGACCTGC-3′.

45 (G) (with dG instead of T in the *EcoRI* restriction site):

5′-GGGATCCTGCTGCCATAGGAA**G**TCTTGATTGGAAAGTCGACCTGC-3′.

45(A) (with dA instead of T in the *EcoR*I restriction site):

5′-GGGATCCTGCTGCCATAGGAA**A**TCTTGATTGGAAAGTCGACCTGC-3′.

45(C) (with dC instead of T in the *EcoRI* restriction site):

5′-GGGATCCTGCTGCCATAGGAA**C**TCTTGATTGGAAAGTCGACCTGC-3′.

Primers:

P1: 5′-GTTGACCTACCCACACCATCCgcaggtcgactttccaatcaa-3′.

(lower case for non-complementary to template part)

P1t: 5′-CY3-GTTGACCTACCCACACCATCC-3′.

P2: 5′-CATAATTACGAGCAATATGAAgggatcctgctgccataggaa-3′.

(lower case for non-complementary to template part)

P2t: 5′-CATAATTACGAGCAATATGAA-3′. All the oligonucleotides were purified by electrophoresis in polyacrylamide gel.

### 2.4. Synthesis of the “Framed Mirror” Products

A general scheme of the “framed mirror” synthesis method is shown in Figure 1.

The synthesis of the “mirror” by cell extracts on the AXT/ddC template was done in a 30 μL reaction mixture containing 50 mM Tris-HCl pH 8.0, 50 mM KCl, 5 mM MgCl_2_, 1 mM EDTA, 1 mM PMSF, 1 mM DTT, 50 µg/mL tRNA, 0.25 mM each of four dNTPs, 400 nM AXT/ddC template, 400 nM P1 primer, and 100 µg protein of nuclear extract. The reaction mixture was incubated for 30 min at 37 °C. The reaction was terminated by adding EDTA up to a concentration of 10 mM. The mixture was then diluted 100 times with reaction buffer. 10 µL of the diluted mixture was supplemented with 2 × DreamTaq Master Mix (Thermo Scientific, Waltham, MA, USA) containing *Taq* polymerase and dNTP, 400 nM primer P2 having homology with the 3′-end of the “mirror”. Full size mirror synthesis by *Taq* polymerase and 1 PCR cycle with *Taq* polymerase (95 °C, 20 s; 95 °C, 20 s; 55 °C, 20 s; 72 °C, 20 s; 72 °C, 5 min) yielded a DNA chain complementary to the “mirror” (single-stranded “framed mirror”). At the next stage, this “framed mirror” was amplified. To this end, asymmetric PCR was carried out in the mixture with an overall volume of 50 μL containing 5 μL of the mixture with the single-stranded “framed mirror”, 500 nM primer P1t labeled withCY3, 5 nM primer P2t, and 25 μL 2 × DreamTaq Master Mix (Thermo Scientific, Toronto, ON, Canada) under the following conditions: 95 °C for 20 s; 25 cycles of 95 °C for 20 s, then 60 °C for 20 s, then 72 °C for 20 s; and 72 °C for 5 min. After this, the second strand of the “framed mirror” was synthesized in 50 μL of the reaction mixture containing 20 μL of the asymmetric PCR product, 25 μL of 2 × DreamTaq Master Mix (Thermo Scientific, Toronto, ON, Canada), and 500 nM primer P2t. The PCR reaction was performed as follows: 95 °C for 20 s; 2 cycles of 95 °C for 20 s, then 55 °C for 20 s, then 72 °C for 20 s; 72 °C for 5 min. At the last stage, the obtained double-stranded “framed mirror” was cut with *EcoRI* at 37 °C for 2 h in the mixture containing 10 μL of the double-stranded “framed mirror”, 2 μL of 10 × *EcoRI* buffer (Thermo Scientific, Toronto, ON, Canada), and 5 U of *EcoRI* (Thermo Scientific, Toronto, ON, Canada).

### 2.5. Electrophoretic Analysis of EcoRI Digestion Products

The products of the *EcoRI* digestion were analyzed by electrophoresis in 20% polyacrylamide gel with Tris-borate buffer pH 7.5, using 12-cm-long glass, at 18 mA current. The electrophoresis ran at 10 °C for 4 h in the dark. The resulting gel was scanned with a Typhoon FLA 9500, and the data were processed with ImageQuant™ v5.2 software.

### 2.6. SSCP Analysis of the Single-Stranded “Framed Mirror”

Single-stranded products of asymmetric PCR were electrophoresed in 20% polyacrylamide gel with Tris-borate buffer, pH 7.5, using 25-cm-long glass, at 9 mA current. The electrophoresis ran for 35 h at 10 °C in the dark. The resulting gel was scanned with a Typhoon FLA 9500, the data were processed with ImageQuant™ v5.2 software and statistically treated using Origin 8.1 program.

## 3. Results

### 3.1. The “Framed Mirror” Method

Earlier, we proposed a “mirror’ method for studying the mutagenic activity of purified DNA polymerases [1]. In the present work, to estimate the mutagenic activity of 8-oxoG in nuclear extracts, we modified a “mirror” method and called it “framed mirror” method. A modification was necessary because DNA-synthesizing activity in cell extracts is rather low. Besides, apart from DNA polymerases, cell extracts contain many other enzymes (nucleases, phosphatases, DNA glycosylases etc.) which can affect the structure of the DNA template under study. The extracts can also contain components able to affect PCR. Taking all this into account, we changed the ‘mirror” method as shown in Figure 1.

At stage 1, a “mirror” is synthesized with primer P1 (see Section 2) on an artificial template containing 8-oxoG in its middle and dideoxycytidine (ddC) at its 3′-end. Primer P1 consisted of two parts. Its 3′-end was complementary to the template and at this end DNA synthesis started. The remaining part was homologous to primer P1t (see Section 2) and served for further amplification of the “mirror” with *Taq* polymerase, irrespective of the template. The ddC located at the 3′-end of the template blocked the PCR synthesis of extra “mirrors”.

At stage 2, the full size “mirror” was synthetized with a site complementary to primer P1t (Figure 1) and was performed one PCR cycle with P2 primer to obtain the “framed mirror”. Simultaneously, the “framed mirror” was converted into the double-stranded form with added sites homologous to primers P1t and P2t for further amplification of the “framed mirror” irrespective of the template. One cycle of PCR with *Taq* polymerase yielded a double-stranded copy of the “framed mirror” (double-stranded “framed mirror”) that could be amplified by PCR with primers Pt1 and Pt2 (Figure 1) non-homologous to the initial template.

synthesis of the “mirror” on a template containing 8-oxoG.Elongation of the “mirror” to obtain full-size “framed mirror” and its conversion into the double-stranded form.asymmetric PCR of the double-stranded “framed mirror”.filling in products of the asymmetric PCR to obtain double-stranded forms.restriction digestion of the double-stranded DNA fragments with *EcoRI*.electrophoretic analysis of the DNA fragments digested by *EcoRI* in polyacrylamide gel.

Elongation in the “mirror” fragments was necessary because cell extracts could not always form full-size copies of the template. Still, the size of most fragments formed in cell nuclear extracts showed that lesions of the template were mainly bypassed, and their filling in with *Taq* polymerase allowed to identify the nucleotide incorporated in the extracts opposite 8-oxoG.

At stage 3, asymmetric PCR with the double-stranded “framed mirror” and P1t primer was performed. This primer was not homologous to the template, and it was the only primer with a Cy3 label at its 5′-end. This PCR yielded Cy3-labeled single-stranded DNA fragments representing copies of the double-stranded “framed mirror”. The reaction did not form copies of the template itself due to the absence of a site homologous to P1t primer and the presence of the ddC-block at the 3′‑end of the template. Additionally, we checked the presence of other mutations that could arise under the action of cell extracts (except the incorporation dA opposite 8-oxoG) using the SSCP method.

At stage 4, to measure the mutagenic potential in extracts more accurately, the single-stranded fragments were filled in with *Taq* polymerase and P2t primer to create double-stranded forms (Figure 1).

At stage 5, double-strand “framed mirror” was treated with *EcoRI* to reveal the mutation in the replication over the lesion.

At stage 6, the products of the *EcoRI* treatment were analyzed in non-denaturing polyacrylamide gel.

### 3.2. Study of the Mutagenic Potential of 8-Oxog in Organs of Adult Mice and Their Embryos

The developed by us “framed mirror” method was used in the present work to determine the frequency of incorporation in the growing DNA chain of the incorrect dA or correct dC opposite 8-oxoG by nuclear extracts of different mouse organs. The initial template was designed so that the incorporation of correct dC opposite 8-oxoG would give a 87 bp PCR product lacking *EcoRI* sites. On the contrary, the incorporation of incorrect dA at the same position should give a PCR product that could be split with *EcoRI* into two fragments, 40 and 47 bp long. However, since the PCR product was labeled with Cy3 at one end, only 47 bp long fragment was visible on the electrophoregram. Figure 2 represents an example of electrophoretic separation of double-strand products synthesized in nuclear extracts of five different mouse organs and treated with *EcoRI*.

As seen in Figure 2, treatment with *EcoRI* gave in all cases two DNA fragments: a 87 bp long fragment due to the incorporation of correct dC, and 47 bp long fragment resulting from the incorporation of mutant dA. Numeric values of the mutagenic potential for each extract were derived from the band intensities.

Figure 3 shows averaged values of the 8-oxoG mutagenic potential in different mouse organs. One can see that this potential in nuclei of testes and the brain (16–18%) is higher than in kidneys, liver and mammary gland (8–13%). Similar experiments with cell nuclear extracts of mouse embryos showed that the mutagenic potential in these extracts was markedly (–25%) higher than in organs of adult mice. Also, it was roughly the same for embryos of different age (12–16 days).

The proposed method can be successfully used not to obtain absolute values, but to compare the mutagenic potential of 8-oxoG in different types of cells. For example, the mutagenic activity of 8-oxoG in the extracts of nuclei of breast cancer cells is significantly higher compared to extracts of the nuclei of normal breast cells (Figure 3).

### 3.3. SSCP Analysis of the Single-Strand “Framed Mirror”

The “framed mirror” method developed by us allows to not only determine the frequency of dA incorporation opposite 8-oxoG in cell nuclear extracts, but also to test for the presence or absence in mammalian cells of factors capable of generating other mutations during translesion synthesis. Their presence should change the banding pattern of the electrophoresed single-strand “framed mirror” obtained by asymmetric PCR at stage 3 of the method. Figure 4 exemplifies such an electrophoresis of samples obtained in analysis of different mouse organs.

It can be seen that patterns of bands 5–8 correspond to only the patterns of *Taq* polymerase synthesis on control templates 45 °C and 45 A. Extra bands due to possible deletions or insertions were not observed. This, together with a detailed analysis of band intensities, showed that the only type of mutation that could be generated by 8-oxoG within the genome was the incorporation of incorrect dA opposite this lesion.

## 4. Discussion

There are some reports that DNA replication is not only coordinated with cell proliferation but is also regulated uniquely in some cell types and organs. The differential regulation of DNA synthesis requires an association between DNA replication and differentiation [16]. These data about association suggest that sets of enzymes involved in the DNA synthesis during differentiation and proliferation are different in different cell types. Therefore, the mutagenic potential of different DNA lesions may depend on the cell type and developmental stage of an organism. However, this type of studies is practically absent. Also, there are no data for the frequency of mutant dA incorporation opposite 8-oxoG instead of correct dC in mammalian cells.

In the present work, we used a new (“framed mirror”) method to study the mutagenic potential of 8-oxoG lesion in different organs of mouse embryos and adult mice. To this end, we used cell nuclei extracts instead of purified enzymes. We believed that such a system was closer to the situation in vivo. The informativeness and efficiency of this system was demonstrated in a number of experiments. For example, earlier we used cell homogenates to reveal alterations in the DNA synthesis and repair during the development of loach *Misgurnus fossilis* [17]. Analysis of sea urchin embryo extracts [17] allowed the authors to reveal the temporal heterogeneity of DNA repair processes in animal development.

The data obtained in the present work showed that the mutagenic potential of 8-oxoG was different in mouse embryos and organs of adult mice. It is widely accepted that mutation frequency is significantly associated with proliferative activity. A relatively high mutagenic potential in nuclear extracts of embryos can be therefore explained by the intense proliferation in embryonic development. In organs of adult mice, the mutagenic potential was the highest in extracts of the brain and testes. The high potential in cell nuclear extracts of testes might be also explained by intense proliferation of cells and the presence of active replicative DNA polymerases that often incorporate dA opposite 8-oxoG within a “mirror” template. However, the relatively high incorporation of incorrect dA in cell extracts of the brain poorly agrees with the accepted view that cells of the brain practically do not proliferate and contain quantitatively prevalent DNA polymerase beta that bypasses 8-oxoG lesion relatively correctly [18,19].

It can be suggested that DNA polymerase activity in the brain differs from that in other organs. In particular, we showed that, in contrast to most other organs, in extracts of mouse brain cells t-stop after incorrectly inserted dT could be overpassed by DNA polymerase iota [20].

Another interesting result of this work is a higher mutagenic potential of 8‑oxoG in mouse embryos compared to organs of adult animals. This suggests that mutagenic activity of DNA lesions is changed during ontogenesis. This finding well corresponds to our earlier data. Also, analysis of incorrect activity of error-prone DNA polymerase iota in *M. musculus* ontogeny demonstrated considerable changes in its activity, which was the highest during the prenatal development of most organs and decreased in the adult body [21].

Using our proposed method, we also showed that the mutagenic activity of 8-oxoG in breast tumor cells is significantly higher than in normal breast cells. This fact is compatible with the well-known concept of a high level of mutagenesis in cancer cells.

## Figures and Tables

**Figure 1 mps-03-00003-f001:**
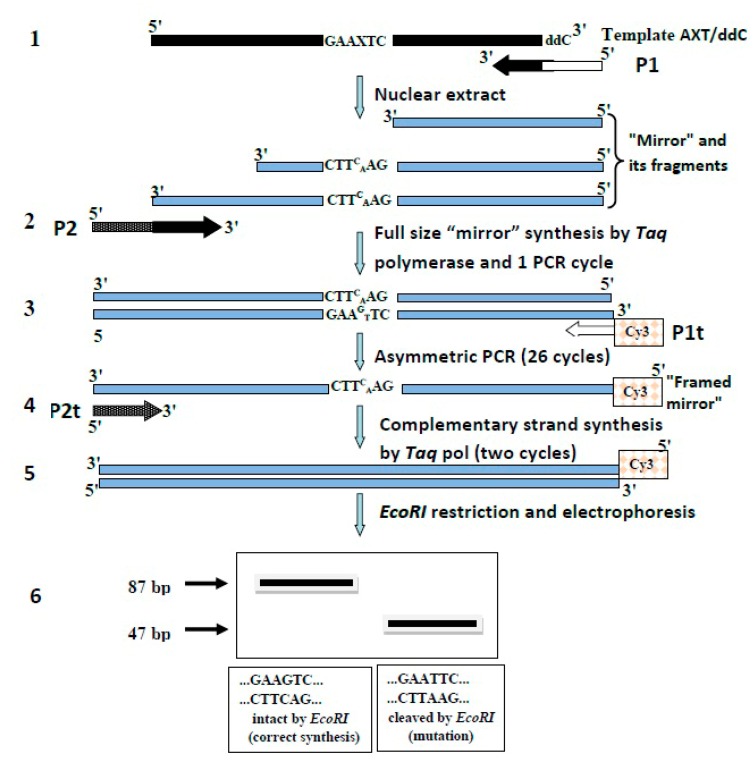
Scheme of the “framed mirror” method.

**Figure 2 mps-03-00003-f002:**
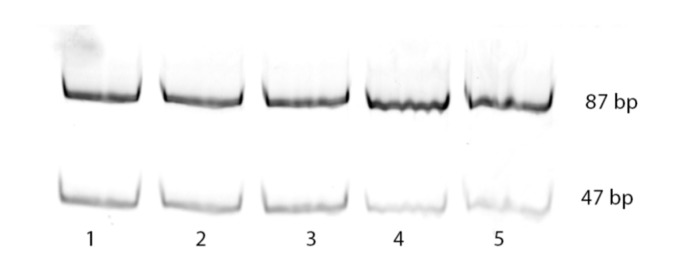
Electrophoregram of the double-stranded products obtained in the extracts of cell nuclei and digested with *EcoRI*: 1—testis, 2—liver, 3—brain, 4—kidney, 5—breast. The percent of 47 bp band was taken as the mutagenic potential of 8-oxoG.

**Figure 3 mps-03-00003-f003:**
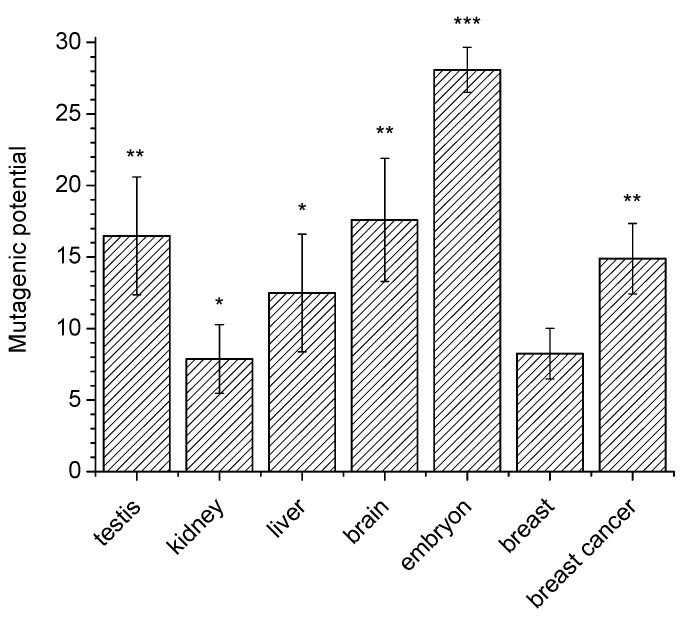
Values of the 8-oxoG mutagenic potential (percentage of the incorporated mutant dA relative to all incorporated nucleotides) measured by the “framed mirror” method in nuclear extracts of various mouse tissues and embryos. *p* Values relative to breast were: * statistically insignificant; ** *p* < 0.02; *** *p* < 0.001 (averaged from at least three experiments).

**Figure 4 mps-03-00003-f004:**
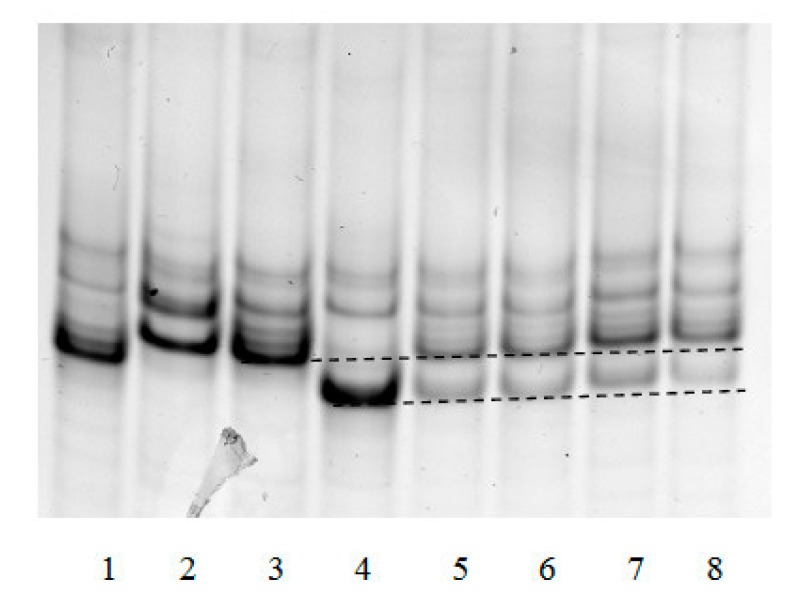
Separation in non-denaturing polyacrylamide gel of the products synthesized by *Taq* polymerase on control templates (1–4) (see Section 2), and in nuclear extracts of different organs on the template containing 8-oxoG (AXT/ddC) (5–8). Templates: 1—45T, 2—45G, 3—45C, 4—45A; organs: 5—the brain, 6—testes, 7—liver, and 8—breast.

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
