# Peer review of "Estimation of the Mutagenic Potential of 8-Oxog in Nuclear Extracts of Mouse Cells Using the “Framed Mirror” Method"

_mps, 2020, doi:10.3390/mps3010003_

Round 1
Reviewer 1 Report
In the present manuscript Gening et al., describe a modified method for estimation of mutagenic potential of 8-oxoG in nuclear extracts derived from mouse organs.
The present work is an improvement of their previously published "mirror" method.
I think the manuscript is well written and the method interesting and useful for the field
I have only a minor point that I would suggest the authors to address.
It would be indeed inshightful if they were able to generate a positive and/or negative control to add in figure 2 and 3. For example a lysate from a cell line which have an increased mutagenic potential and compared it to a normal non cancerous cell line.
Author Response
Point1
In the present manuscript Gening et al., describe a modified method for estimation of mutagenic potential of 8-oxoG in nuclear extracts derived from mouse organs.
The present work is an improvement of their previously published "mirror" method.
I think the manuscript is well written and the method interesting and useful for the field
I have only a minor point that I would suggest the authors to address.
It would be indeed inshightful if they were able to generate a positive and/or negative control to add in figure 2 and 3. For example a lysate from a cell line which have an increased mutagenic potential and compared it to a normal non cancerous cell line.
Response 1
We added data obtained in experiments with nuclear extracts of mice breast cancer cells. We added an additional sentence in the text of the article (starting from line 241): The proposed method can be successfully used not to obtain absolute values, but to compare the mutagenic potential of 8-oxoG in different types of cells. For example, the mutagenic activity of 8-oxoG in the extracts of nuclei of breast cancer cells is significantly higher compared to extracts of the nuclei of normal breast cells (Figure 3)”.
In addition, we replaced the incorrect word “filling” with “elongation” (line 199).
Reviewer 2 Report
Authors developed their method which had been introduced in a previous publication. They applied primers containing extra flank fragments for in vitro DNA replication with nuclear extracts. They estimated that the development would eliminate or at least to decrease the negative effect from nucleases in the nuclear extract. The innovation is interesting. However, a serious concern rise with the fragments synthesized by extract. This may not be an issue when the assay was performed with purified protein since the extension would be fulfiled.
As authors mentioned in their manuscript, a lot of "Mirror" products would present as short products instead of full length ones due to the low frequency of polymerases and the interruption from regulatory machinery. These short products lead to an ambiguous estimation of mutation frequency. My reason is as follows:
Since an extra one cycle PCR with a primer attaching to the 3'-end was performed to synthesize the complimentary strand of "Mirrors", only full length "Mirrors" goes into next step. The information in short products is abandoned. The information in short products can not ignored because both the 8-oxoG lesion and mis-matches induce DNA repair which will inhibit the extension.
Authors mentioned that they applied an "filling" process to form full-size copies. I am confused by this description. Did authors extend the short products to full-length so that the gap was filled? To my best understanding, authors synthesized the complimentary strand with P2 in the "filling" step. Therefore, the issue of abandoned information, which was mentioned in the last paragraph, was not resolved.
If authors indeed fill the short products to full length ones, another concern may rise. In this case, the extension of short products which do not touch the lesion site will lead to an over estimation of correct incorporation.
I would like to suggest authors to investigate the yield of full length extension by using P32 labeled P1. If most products are short ones, a precise investigation on these short DNA is required.
Author Response
Point 1
Since an extra one cycle PCR with a primer attaching to the 3'-end was performed to synthesize the complimentary strand of "Mirrors", only full length "Mirrors" goes into next step. The information in short products is abandoned. The information in short products can not ignored because both the 8-oxoG lesion and mis-matches induce DNA repair which will inhibit the extension.
Response 1
We are especially grateful to reviewer 2 for his remark concerning the description of the mirror synthesis. In fact, in stage 2, we first performed a full size mirror synthesis and then 1 PCR cycle. In this regard, we added corrections to the figure and to the text in two places (line 142 in the updated manuscript and line 185).
Point 2
Since an extra one cycle PCR with a primer attaching to the 3'-end was performed to synthesize the complimentary strand of "Mirrors", only full length "Mirrors" goes into next step. The information in short products is abandoned. The information in short products can not ignored because both the 8-oxoG lesion and mis-matches induce DNA repair which will inhibit the extension.
Authors mentioned that they applied an "filling" process to form full-size copies. I am confused by this description. Did authors extend the short products to full-length so that the gap was filled? To my best understanding, authors synthesized the complimentary strand with P2 in the "filling" step. Therefore, the issue of abandoned information, which was mentioned in the last paragraph, was not resolved.
If authors indeed fill the short products to full length ones, another concern may rise. In this case, the extension of short products which do not touch the lesion site will lead to an over estimation of correct incorporation.
Response 2
Generally, we agree with the remark. However, our control experiments (not described in the article) showed that in more than 80-90% of cases, shortened products overcome the region containing 8-oxoG. We believe that our method can be successfully used not to obtain absolute values, but to compare the mutagenic potential of different organs. The most adequate results can be achieved by studying the same type of cells, but under different conditions. To confirm this, we added data obtained with nuclear extracts of mouse breast cancer cells in figure 3. A proposal was made in the text of the article (line 241): The proposed method can be successfully used not to obtain absolute values, but to compare the mutagenic potential of 8-oxoG in different types of cells. For example, the mutagenic activity of 8-oxoG in the extracts of nuclei of breast cancer cells is significantly higher compared to extracts of the nuclei of normal breast cells (Figure 3).” In addition, we replaced the incorrect word “filling” with “elongation” in line 199.
Reviewer 3 Report
Overall impression.
The manuscript describes an update of the “mirror” method (previously designed by the authors) to study DNA lesion bypass with cell extracts instead of purified DNA polymerases (pols). The design looks intelligent and robust, and the performance of the assay is demonstrated, though it would be good to see control experiments e.g., with a template without terminal ddC, without extract, with partially inactivated extract, or with isolated pols at the first step. It would be of interest to discuss how the bypass of 8-oxoG by extracts compares to bypass by isolated DNA pols. In the future, it would be interesting to compare extracts made from organs of mice deficient in certain translesion DNA synthesis pols to reveal the impact of individual pols in the complex mixtures.
The text should be corrected for the proper use of terminology. English requires some polishing.
Detailed comments.
Abstract.
Lines 10-12. The first sentence is long and difficult to follow.
…14. Clarify what is 8-oxoG “inside”.
…15. Maybe it will be useful to state what part of primers was non-complementary.
…18-20. The authors estimated the frequency of the particular type of 8-oxoG bypass events by cell extracts, not mutation frequency. In the next statement, the "mutagenic potential" is an exaggeration.
…25. Remove “…is such …as”.
…30. Substitute “entire” by “all”. Correct “…with different frequencies AND along…”.
…31, and 259. What is “mutant dA”?
…31-32. The cited paper deals with pol delta and pol lambda. The authors should better cite a review where the data for other pols are summarized.
…32-34. A review on the mechanism of 8oxo-G mutagenesis should be cited. The mechanism was elucidated way before the mentioned paper, see for example J Bacteriol. 1992 Oct; 174(20): 6321–6325
…35. We suggest removing the first six words.
…37-38. A more accurate description is desired, and additional papers should be cited, e.g., EMBO J. 2016, 35(18): 2045–2059.
…45. Clarify what are “other mutations”.
…49-52. Edit this phrase to remove “…was used…carried out…was performed…”
…52-55. Clarify what it means "not separated". Again, the misincorporation of a nucleotide does not equal to mutation.
…65. With the first mention, it is desirable to explain what is “SSCP”
…71. Do the authors mean “fidelity of DNA synthesis”?
…72. Cite some review on the subject?
…73. Elaborate on “…knocked down cells…”
Materials and Methods.
…80;275. Should be "repair…". To what “they” refers to – to nucleases, pols, or cell extracts?
…84. Maybe …we modified the "mirror" method?
…108. “Poured” or “pooled”?
…113. “firm”, not “firms”
Results.
...164, 231-233, Fig 3. The authors should provide more details on how the data were statistically treated.
…179. “matrix” or “template”?
…180. Can this ddC be removed immediately by nucleases in the extract?
…168-169; 272. Homogenates of nuclear extracts? Again, maybe …we modified…
…172. Can the authors be more specific about the effect of all these proteins on “…the structure of the DNA…”
…181, 196. The use of the word "…filled..” is misleading. One can fill the gap, but the gap here is not present. Fig.1 says, "second strand synthesis." Or, maybe, complementary strand?
…205-206. The sentence could be simplified.
We suggest that the authors shorten the discussion and concentrate on technical aspects.
Discussion.
…262-263. Can the authors be more specific and also add appropriate references?
…264. Crosslink between replication and differentiation? “There data” – what data?
…265-266. The message is misleading. The differences between samples are pretty small, and the reasons for the variation are not studied at all.
...293-296. We recommend avoiding generalization of the results with loach. Defects of mismatch repair cause early-onset cancer, Nat Genet. 2015;47(3):257-62
...289-281. We would recommend removing this statement; it goes beyond the scope of the current methodological work.
...278-288. The term "mutagenic potential is repeated six times in the short paragraph.
...279-280. Add appropriate references.
...299-301. We recommend to tone down the statements about the broad significance of the current results. The authors offer an update on methodology and studied bypass of 8-oxoG at only one site in the only one DNA sequence context. They did not investigate how this bypass relates to mutation rates in genomes of different tissues and cells.
Author Response
Please see the attachment. Our answers are in red.

Round 2
Reviewer 2 Report
My concerns have been addressed. However, there are a few minor issues that need to be fixed before publication.
There is an extra bar under the top stand in step 3 in Figure 1. I assume that the P>0.05 was an typo in the caption of Figure 3.Please proofread the manuscript carefully.
Author Response
Point 1
There is an extra bar under the top stand in step 3 in Figure 1.
Response 1
The correction is done.
Point 2
I assume that the P>0.05 was an typo in the caption of Figure 3.
Response 2
The correction is done.